# *Fusarium* and Hazelnut: A Story of Twists and Turns

Beata Zimowska [1], Agnieszka Ludwiczuk [2], Gelsomina Manganiello [3], Krzysztof Wojtanowski [2], Izabela Kot [1], Alessia Staropoli [3], Francesco Vinale [4] and Rosario Nicoletti [3,5,*]

1   Department of Plant Protection, University of Life Sciences, 20-400 Lublin, Poland; beata.zimowska@up.lublin.pl (B.Z.); izabela.kot@up.lublin.pl (I.K.)
2   Department of Pharmacognosy with the Medicinal Plant Garden, Medical University of Lublin, 20-439 Lublin, Poland; agnieszka.ludwiczuk@umlub.pl (A.L.)
3   Department of Agricultural Sciences, University of Naples 'Federico II', 80055 Portici, Italy; gelsomina.manganiello@unina.it (G.M.); alessia.staropoli@unina.it (A.S.)
4   Department of Veterinary Medicine and Animal Production, University of Naples 'Federico II', 80138 Naples, Italy; fr.vinale@unina.it
5   Council for Agricultural Research and Economics, Research Center for Olive, Fruit and Citrus Crops, 81100 Caserta, Italy
*   Correspondence: rosario.nicoletti@crea.gov.it

**Abstract:** In recent years, the number of reports of *Fusarium* in association with hazelnut (*Corylus avellana*) has been increasing worldwide, related to both pathogenic aptitude and endophytic occurrence. However, the assessment of the real ecological role and relevance to plant health of these fungi has been impaired by uncertainty in species identification, deriving from both the evolving taxonomic structure of the genus and an inaccurate use of molecular markers. In this paper, the characterization of two hazelnut endophytic strains isolated in Poland is reported with reference to their secondary metabolite profiles and interactions with pests and pathogens. Our results are indicative of a possible role of these strains in defensive mutualism which could be related to the production of several bioactive compounds, especially cyclohexadepsipeptides of the enniatin family. At the same time, these biochemical properties create some concern for the possible mycotoxin contamination of hazelnut products.

**Keywords:** defensive mutualism; endophytic fungi; enniatins; *Fusarium citricola* species complex; mycotoxins; phylogenetic analysis

## 1. Introduction

Within the several fungal diseases reported among the adversities of hazelnut (*Corylus avellana* L.), those affecting fruits are of major concern due to their implications for product storage and possible mycotoxin contamination. Indeed, many fungi may have an economic impact in terms of the quality of fresh and confectionery products, making it necessary to achieve adequate control in both pre- and post-harvest fruit management [1]. While species of *Penicillium* and *Aspergillus* basically have impacts during storage and processing, other fungi start their infections in the field; hence, they can be monitored and more timely managed. This is the case for species of *Fusarium*, which establish various kinds of ecological interactions with the host plants and their associated organisms and are notorious mycotoxin producers [2].

Despite their widespread occurrence, infections of hazelnut fruits by *Fusarium* have only started being documented quite recently; however, some reports limit identification to the genus level [3–7], which is not very informative in terms of the possible phytopathological or ecological significance. In other cases, identifications were performed at the species level, but these data are not always entirely reliable because of both the evolving taxonomic structuring of *Fusarium*, still based on several 'species complexes' awaiting to be resolved [8,9], and the need to use an appropriate set of DNA markers. In Chile, isolates from brown-grayish spots on the nuts were identified as *F. sporotrichioides* after sequencing

the internal transcribed spacers of rDNA (ITS) [10]; although of widespread use in fungal taxonomy, this marker is notoriously insufficient to correctly identify *Fusarium* spp. [9,11]. In fact, an updated blast in GenBank of the published sequences of these strains showed that one of them (code MF629827) matches better with *F. avenaceum* and *F. lateritium*, leaning for a more heterogeneous species assortment. Two isolates from moldy fruits collected in the field in Oregon were classified as *F. lateritium* and *F. culmorum* through ITS and translational elongation factor (*tef-1*) sequencing [12], the latter marker being more reliable for species identification in *Fusarium*; however, an updated blast in GenBank showed that both isolates match better with *F. lateritium* and a more recently described species (*F. citricola*), which is sister to the *F. tricinctum* species complex (FTSC) [13]. Indeed, the discovery of new *Fusarium* species is ongoing, and recent findings call for taxonomic adjustments even within the *F. lateritium* species complex (FLSC) [14].

*Fusarium lateritium* had previously been reported as the agent of gray necrosis of hazelnuts (NGN) in Latium, central Italy [15]. In that case, species identification was based on a complete set of DNA markers, and a phylogenetic analysis documented a certain heterogeneity of the bulk of isolates ascribed to this species at that time, with a clear separation between European and non-European isolates, regardless of the host [16]. More recently, the examination of isolates from hazelnuts affected by NGN in the same area confirmed morphological similarity with *F. lateritium*; however, the draft genome sequencing of one of these isolates demonstrated its closer proximity to *F. tricinctum* [17], introducing uncertainty about the correct identification of both the hazelnut isolates and those with DNA sequences deposited in GenBank as *F. lateritium*, which have been used as references over time. On the other hand, the possible misidentification of strains classified as *F. lateritium*, but actually more closely related to *F. tricinctum*, had already been pointed out in an earlier phylogenetic study [18].

Data concerning endophytic occurrence of *Fusarium* in hazelnut are even more limited and refer to a few studies carried out in Iran and Turkey in which isolates from several plant parts were ascribed to the species *F. equiseti, F. fujikuroi, F. graminearum, F. oxysporum, F. proliferatum,* and *F. tricinctum* based on morphological features or ITS sequencing [19]. Nevertheless, *Fusarium* spp. have a widespread occurrence as endophytes in tree crops [20–26]; their association with plants could be attributable either to interception during the latency stage of the disease cycle or to a real endophytic settlement with implications in defensive mutualism against pests and pathogens. The latter role could be mediated by bioactive secondary metabolites, which might also lead to fruit contamination with mycotoxins.

Indeed, the accumulation of additional data from other geographical areas is appropriate for disentangling the taxonomic puzzle of *Fusarium* associates of *C. avellana*, as well as for increasing our knowledge on the real impact of these fungi on plant health and the safety of hazelnut products. Here, we report the identification of two endophytic strains collected from branches of asymptomatic hazelnut trees thriving in a forest area in south-eastern Poland, their metabolomic profiles, as well as effects against plant pathogenic fungi and a model insect.

## 2. Materials and Methods

### 2.1. Isolation and Morphological Characterization

Within the cooperative research on endophytic fungi of *C. avellana* in progress at our laboratories [27], two *Fusarium* isolates (Hzn1 and Hzn5) were recovered from asymptomatic tissues of secondary branches of hazelnut plants collected in the Lublin voivodeship, south-eastern Poland (51.235598° N, 22.385680° E). Branch segments of about 4 cm in length were cut and surface-sterilized by consecutive immersions in 70% ethanol for 1 min, 3% sodium hypochlorite for 3 min, and 70% ethanol again for 1 min, and finally washed three times in sterile distilled water for 1 min. After removing bark with a sterile scalpel, the sterilized samples were cut into five pieces of 8 mm and aseptically transferred to Petri dishes containing potato dextrose agar (PDA: Difco, Detroit, MI, USA). The dishes were incubated at 25 ± 1 °C and the emerging endophytic fungi were transferred onto fresh

PDA. From these plates, single-spore subcultures on PDA were prepared for each isolate prior to the identification procedure.

To investigate the morphological characteristics, the *Fusarium* strains were individually cultured on PDA and synthetic nutrient agar (SNA, made from ingredients in the laboratory) in darkness at 25 ± 1 °C. After 10 days, observations of phenotypes on PDA were carried out considering colony diameter, margins, pigmentation, and averse and reverse colors [28]. Micromorphological features concerning the formation of conidiogenous cells, conidia, and chlamydospores were inspected and photographed using a light microscope (Nikon Eclipse Ni-U, Tokyo, Japan).

*2.2. Biomolecular Markers and Phylogenetic Analysis*

Morphological features were integrated with the sequencing of relevant DNA markers, namely ITS, *tef-1*, the RNA polymerase II (*rpb2*), and β-tubulin (*tub*) genes. One hundred milligrams of mycelium collected from PDA cultures of isolates Hzn1 and Hzn5 were ground in liquid nitrogen using a mortar and a pestle. The mycelial powder was transferred to sterile 2 mL tubes and DNA extraction was performed using the E.Z.N.A.® Plant & Fungal kit (Omega Bio-tek, Norcross, GA, USA), according to the manufacturer's protocol. The extracted DNA was dissolved in 50 μL water (molecular biology grade), and its concentration and purity were checked through a DS-11 spectrophotometer (DeNovix Inc., Wilmington, DE, USA). PCR amplification was performed in 30 μL of DreamTaq Green PCR Master Mix (Thermo Scientific, Waltham, MA, USA) with 2 μL of genomic DNA and 1 μL of each primer (Table 1). The reaction parameters were initial denaturation at 95 °C for 5 min, followed by 35 cycles at 95 °C for 30 s, 58 °C for 30 s, and 72 °C for 45 s, with a final extension step at 72 °C for 10 min. PCR products were then run on 1.5% agarose gel stained with SYBR Safe DNA Gel Stain (Thermo Scientific) along with a 1 kb DNA ladder (A&A, Gdansk, Poland) to estimate the size of the amplified bands, and purified using magnetic beads (Quantabio, Beverly, MA, USA). Sanger sequencing of the purified PCR products was performed using the BigDye Terminator v3.1 cycle sequencing kit (Applied Biosystems, Waltham, MA, USA). Sequences were generated on an Applied Biosystems 3130XL Genetic Analyzer, optimized, and corrected manually when necessary. The obtained sequences were subjected to individual blast searches in GenBank for a preliminary identification based on sequence homology.

**Table 1.** Primers used for Hzn1 and Hzn5 housekeeping gene amplifications.

| Primer | Sequence 3′–5′ | Reference |
|---|---|---|
| ITS1-F | CTTGGTCATTTAGAGGAAGTAA | [29] |
| ITS4 | TCCTCCGCTTATTGATATGC | [30] |
| bRPB2-6F | TGGGGYATGGTNTGYCCYGC | [31] |
| bRPB2-7R | GAYTGRTTRTGRTCRGGGAAVGG | |
| EF1-1018F | GAYTTCATCAAGAACATGAT | [32] |
| EF1-1620R | GACGTTGAADCCRACRTTGTC | |
| T1 | AACATGCGTGAGATTGTAAGT | [33] |
| Bt2b | ACCCTCAGTGTAGTGACCCTTGGC | [34] |

A phylogenetic analysis was carried out based on the concatenated *rpb2* and *tef-1* sequences, including reference strains of all of the species recognized within the *Fusarium citricola* species complex (FCCSC), four selected strains of *F. tricinctum*, and a strain of *F. lateritium* which was considered a genuine representative of this species in a recent comprehensive taxonomic study on *Fusarium* [35]. Moreover, two strains identified as *Fusarium* sp. were also included, since they both had *rpb2* and *tef-1* sequences deposited in GenBank and had high homology with our hazelnut isolates (Table 2). Individual gene sequences were validated through alignment using ClustalW 2.1 in MEGA X with the default alignment parameters [36] and checked manually. Sequences of *rpb2* and *tef-1* were then combined for multi-locus sequencing analysis (MLSA). Concatenated sequences

were aligned and checked for alignment errors in the same manner as the individual gene sequences. The evolutionary history was inferred using the maximum likelihood method and Tamura-Nei model [37]. Initial trees for the heuristic search were obtained automatically by applying Neighbor-Joining and BioNJ algorithms to a matrix of pairwise distances estimated using the Tamura-Nei model and selecting the topology with the superior log likelihood value.

**Table 2.** DNA sequences of reference strains used in phylogenetic analysis.

| Isolate | Species | Origin | *tef-1* | *rpb2* |
|---|---|---|---|---|
| UBOCC-A-109005 | *F. aconidiale* | *Triticum aestivum*, France | MZ078246 | MZ078218 |
| MFLUCC 16-0526 | *F. celtidicola* | *Celtis australis*, Italy | ON745620 | ON759296 |
| CPC 27067 | *F. citricola* | *Citrus limon*, Italy | LT746194 | LT746307 |
| CPC 27069 | *F. citricola* | *Citrus sinensis*, Italy | LT746195 | LT746308 |
| CPC 27709 | *F. citricola* | *C. sinensis*, Italy | LT746196 | LT746309 |
| CPC 27805 | *F. citricola* | *Citrus reticulata*, Italy | LT746197 | LT746310 |
| CPC 27813 | *F. citricola* | *C. reticulata*, Italy | LT746198 | LT746311 |
| UBOCC-A-101147 | *F. juglandicola* | *Juglans regia*, France | MZ078244 | MZ078216 |
| UBOCC-A-102014 | *F. juglandicola* | *J. regia*, France | MZ078245 | MZ078217 |
| UBOCC-A-119001 | *F. juglandicola* | *J. regia*, France | MZ078243 | MZ078215 |
| NRRL 13622 | *F. lateritium* | *Ulmus americana*, USA | AY707173 | JX171571 |
| CPC 26403 | *F. salinense* | *C. sinensis*, Italy | LT746191 | LT746304 |
| CPC 26457 | *F. salinense* | *C. sinensis*, Italy | LT746192 | LT746305 |
| CPC 26973 | *F. salinense* | *C. sinensis*, Italy | LT746193 | LT746306 |
| F1544 | *F. tricinctum* | *Triticum turgidum*, Italy | OL964791 | OL658768 |
| LC0453 | *F. tricinctum* | *Hosta* sp., China | MW620151 | MW474676 |
| NRRL 25481 | *F. tricinctum* | *T. aestivum*, Germany | OL772833 | MH582357 |
| P325b | *F. tricinctum* | *T. turgidum*, Italy | OL658799 | OL658796 |
| IHEM 28077 | *Fusarium* sp. | bat, Belgium | OU641411 | OU641410 |
| ZLVG.982 | *Fusarium* sp. | *Pinus sylvestris*, Slovenia | OR105858 | OR098304 |

### 2.3. Antagonism against Plant Pathogenic Fungi

The interactions between the isolate Hzn5 and three strains of fungal pathogens which are also reported to infect hazelnut, namely *Botrytis cinerea*, *Colletotrichum gloeosporioides*, and *Diaporthe eres*, from the mycological collection of the Department of Plant Protection of the University of Life Sciences in Lublin, were examined in dual cultures. All of the fungi were preliminarily grown on PDA for 7 days at 25 ± 1 °C. Test plates were prepared on PDA by placing 5 mm diameter mycelial plugs of Hzn5 at the center and two mycelial plugs of each pathogen on the left and right sides at the same distance. The cultures were incubated at 25 ± 1 °C in darkness, and the inhibitory effects were visually recorded after 7 and 14 days.

### 2.4. Bioassays on Aphids

Parthenogenetic colonies of the bird cherry-oat aphid (*Rhopalosiphum padi*) are maintained at the Department of Plant Protection as a laboratory stock on plantlets of wheat (*Triticum aestivum*) in an environmental chamber at 25 ± 1 °C, with a 16:8 light–darkness photoperiod and 60 ± 5% relative humidity. Twenty nymphs from these colonies were placed onto new plantlets at the three-leaf stage. After the establishment of the new colonies, 5 mL of a conidial suspension of the isolate Hzn5 was applied using a hand sprayer. The conidial suspension was prepared by rinsing the surface of SNA cultures of Hzn5 with sterile distilled water mixed with 0.01% Tween 80, adjusting it to a density of $10^5$ conidia $mL^{-1}$ by using a hemocytometer. Control plants were treated at the same time with sterile distilled water mixed with 0.01% Tween 80. Both treatments were applied in triplicates. The number of adults and nymphs was counted daily to follow the colony development over 10 days after the treatment.

At the end of the experiment, 10 adult aphids were collected from the treated plants and plated on PDA after mild surface sterilization (70% ethanol for 1 min, followed by washing in sterile distilled water for 1 min). Moreover, fragments cut from both the leaves and stem under the sheaths were surface disinfected by dipping them in 3% sodium hypochlorite for 2 min, washed three times in sterile distilled water, and plated on PDA to

check for possible colonization by Hzn5 of the inner tissues of the treated plants. The plates were kept in darkness at 25 ± 1 °C; on emergence, the fungal colonies were transferred to pure cultures for morphological identification.

*2.5. Metabolomic Analysis*

Following our previous experience concerning the differential production of secondary metabolites by *Fusarium* strains in different media [38], liquid cultures of isolates Hzn1 and Hzn5 were prepared on Czapek–Dox broth (CDB, Difco) and malt extract broth (MEB, Difco) in 500 mL Erlenmeyer flasks containing 250 mL of substrate. After two weeks of growth in darkness at 25 ± 1 °C, the cultures were filtered with 0.45 μm syringe filters, and the culture filtrates were extracted with an equal volume of chloroform in a separation funnel. The chloroform fractions were dried to remove residual water using anhydrous sodium sulfate, filtered, and concentrated under reduced pressure. They were then redissolved in methanol and qualitatively analyzed by a HPLC/ESI-QTOF-MS system in both positive and negative ion mode with a 6530B accurate-mass QTOF-MS mass spectrometer (Agilent Technologies, Santa Clara, CA, USA) with an ESI-Jet Stream ion source. The Agilent 1260 chromatograph was equipped with a DAD detector, autosampler, binary gradient pump, and column oven. The extract (10 μL) was injected and eluted using a mixture of water with 0.1% formic acid (solvent A) and acetonitrile with 0.1% formic acid (solvent B) as the mobile phase, with the following gradient: 0–45 min, 0–60% B; 45–46 min, 60–95% B; 46–55 min 95% B; the post-time was 10 min. The total time of analysis was 65 min, with a stable flow rate of 0.2 mL $min^{-1}$. ESI-QToF-MS analysis was performed according to the following parameters of the ion source: dual-spray jet stream ESI; positive and negative ion mode; gas ($N_2$) flow rate of 12 L $min^{-1}$; nebulizer pressure of 35 psig; vaporizer temperature of 300 °C; m/z range of 100–1000 mass units, with acquisition Mode Auto MS/MS; collision-induced dissociation (CID) at 10 and 30 eV with an MS scan rate of one spectrum per s and two spectra per cycle; skimmer at 65 V; fragmenter at 140 V; octopole RF Peak of 750 V. The identification of compounds was based on both an in-house database and the MS-DIAL database (http://prime.psc.riken.jp/compms/msdial/main.html, accessed on 5 February 2024).

To check the possible release of secondary metabolites in plants treated with the conidial suspension of the isolate Hzn5, at the end of the experiment, the leaves and stems of wheat plants were cut into small pieces and an amount of 0.3 g was crushed in a mortar together with 3 mL chloroform as the extraction solvent. The extraction was repeated three times, and the crude extract was filtered through a Pasteur pipette packed with celite. The filtered extract was evaporated to afford a residue of 20 mg, which was redissolved in 1 mL methanol and analyzed by LC/MS following the above-described procedure.

## 3. Results

*3.1. Identification and Phylogenetic Analysis*

The colonies of isolates Hzn1 and Hzn5 were very similar in appearance: after a 10-day incubation, they reached a diameter of 45–47 mm on PDA and 78–85 mm on SNA. Aerial mycelium on PDA was abundant and dense, floccose to woolly, and white–cream with a crenate margin. The reverse was salmon-pink, darkening at the center, while no pigmentation was noticed in the medium (Figure 1). In older 30-day cultures on PDA, dark grey sclerotia-like structures were visible. On SNA, macroconidia were abundantly formed in sporodochia, from monophialidic conidiogenous cells, while microconidia were absent. Chlamydospores were formed quickly and abundant, mainly in chains, but also single or paired, smooth-walled, intercalary, and globose–subglobose to pyriform.

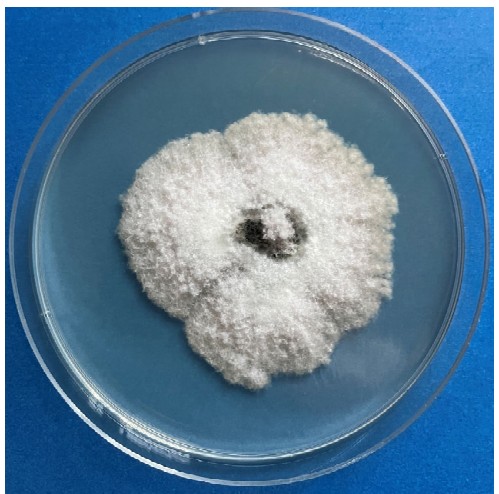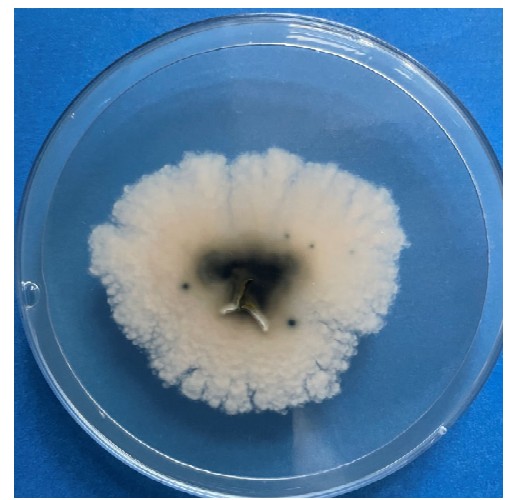

**Figure 1.** PDA culture of isolate Hzn5 (averse at (**left**), reverse at (**right**)).

The obtained sequences of the DNA markers to be considered for biomolecular identification of isolates Hzn1 and Hzn5 were submitted to GenBank, with the following corresponding codes: Hzn1: OR178404 (ITS), PP738981 (*rpb2*), and PP788632 (*tef-1*); Hzn5: OR178403 (ITS), PP738982 (*rpb2*), PP788631 (*tef-1*), and PP791104 (*Tub*).

While confirming that isolates Hzn1 and Hzn5 are conspecific, the results of the blasts in GenBank displayed an inconsistency in their closest matches with the sequences available in this repository (Table 3), indicating uncertainty in species identification when merely based on DNA sequence homology. Therefore, a phylogenetic analysis was performed based on concatenated *rpb2* and *tef-1* sequences, which were considered more reliable markers for correct taxonomic identification with reference to the sequences of members of the FCCSC available in GenBank. This analysis unequivocally demonstrated that the two hazelnut endophytic isolates belong to this species complex, with a closer proximity to *F. celtidicola* (Figure 2). The additional strains included in the analysis because of high sequence homology with our isolates were also determined to belong to the FCCSC; in fact, the Belgian strain IHEM 28077 clustered together with the single available strain of *F. aconidiale*, while the Slovenian strain ZLVG.982 was positioned on an independent branch related to *F. salinense*.

**Table 3.** Results of GenBank blasts of DNA sequences of isolates Hzn1 and Hzn5.

|  | ITS | *tef-1* | *rpb2* | *Tub* |
|---|---|---|---|---|
| Hzn1 | OP699807 *F. juglandicola* id 99.82, qc 99% | OP715604 *F. juglandicola* MZ191070 *F. lateritium* * id 99.24, qc 99% | ON759296 *F. celtidicola* id 99.73, qc 99% | |
| Hzn5 | OP699807 *F. juglandicola* id 99.82, qc 99% | OP715604 *F. juglandicola* MZ191070 *F. lateritium* * id 99.24, qc 99% | ON759296 *F. celtidicola* OL690434 *F. juglandicola* MZ078218 *F. aconidiale* id 99.86, qc 99% | MZ191071 *F. lateritium* * id 100, qc 96% |

id: % identity; qc: query cover; * species identification of these strains is questionable.

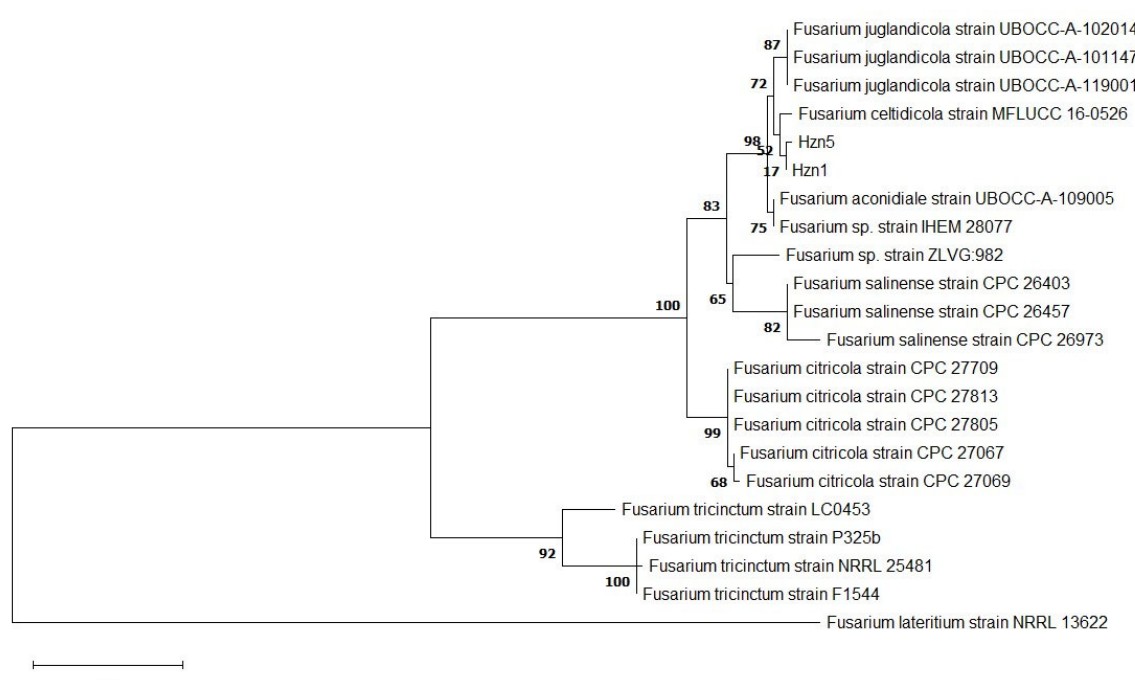

**Figure 2.** Phylogenetic analysis inferred through the maximum likelihood method and Tamura-Nei method, employing concatenated sequences of RNA polymerase II (*rpb2*) and translational elongation factor (*tef-1*) housekeeping genes. The tree with the highest log likelihood (−2918.13), displaying relationships of hazelnut endophytic isolates with species in the FCCSC, *F. tricinctum* and *F. lateritium*, is reported. The percentage of trees in which the associated taxa clustered together is shown next to the branches. This analysis involved 22 nucleotide sequences. The codon positions included were 1st + 2nd + 3rd + Noncoding. There were a total of 1144 positions in the final dataset. Evolutionary analyses were conducted in MEGA X.

### 3.2. Effects against Plant Pathogenic Fungi and Aphids

The growth of the tested plant pathogens was clearly affected in dual cultures with both *Fusarium* strains; the formation of inhibition zones, which persisted over two weeks, was indicative of the release of antibiotic products. The largest inhibition zone was observed in dual cultures with *C. gloeosporioides* (Figure 3). In the absence of contact between the opposing strains, no direct mycelial interactions could be observed.

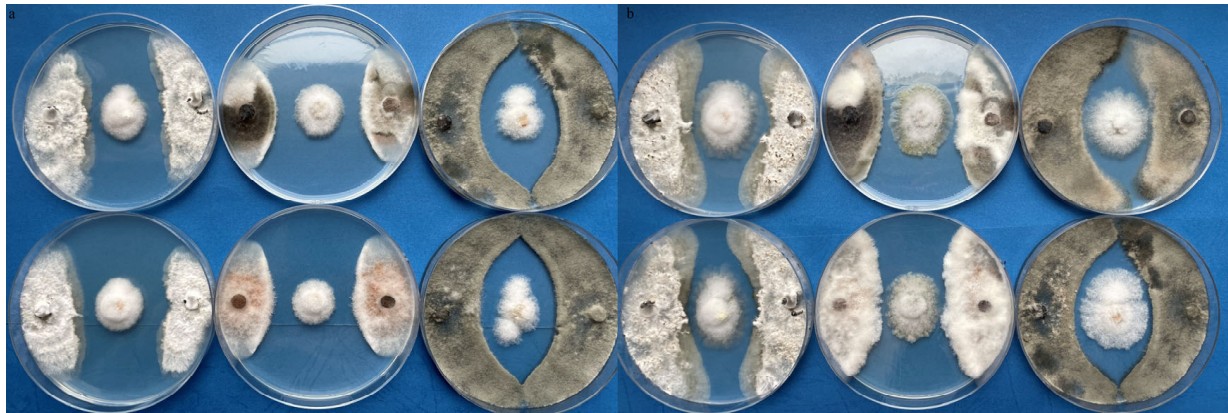

**Figure 3.** Inhibitory effects of isolate Hzn5 against strains of *Diaporthe eres* (**left**), *Colletotrichum gloeosporioides* (**center**), and *Botrytis cinerea* (**right**) after one (**a**) and two (**b**) weeks of co-culture on PDA.

An inhibitory effect was also evident in the development of colonies of *R. padi* on wheat plants. In fact, the number of adult aphids declined at a higher rate on plants treated with the conidial suspension, while the number of nymphs increased at a much lower rate compared to the control plants (Figure 4). Both of these trends are indicative of detrimental effects deriving from the inoculation of the isolate Hzn5, which could be dependent on either its entomopathogenic aptitude, or again the release of toxic products after its eventual settling in the plant tissues. As a matter of fact, the isolate Hzn5 could be re-isolated on PDA from the treated plants, as well as from all of the aphids sampled, demonstrating the ability of this fungus to establish itself as an endophyte in wheat plants and to infect *R. padi*. In the latter respect, it was not possible to assess whether the aphids used for re-isolation were infected during the treatment, or whether they assumed the fungus by sap sucking or through contact with other aphids. Of course, the real modalities have great relevance and deserve to be more deeply investigated in dedicated experiments.

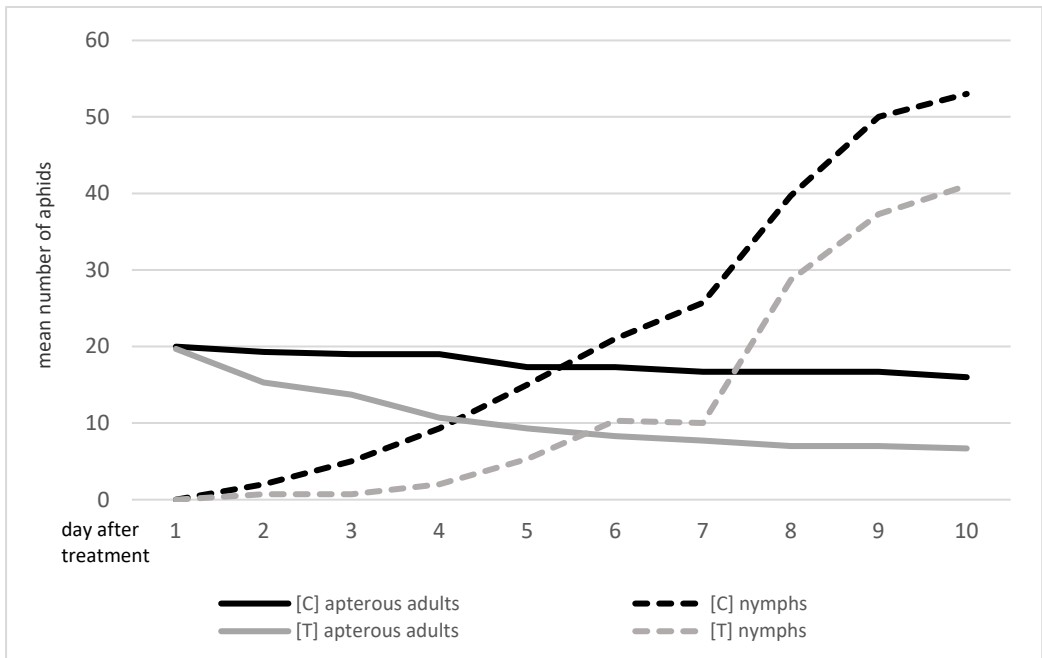

**Figure 4.** Number of adults (bold line) and nymphs (dashed line) in colonies of *Rhopalosiphum padi* treated (T) or not (C) with conidial suspension of isolate Hzn5.

### 3.3. Metabolomic Analysis

The conspecificity of isolates Hzn1 and Hzn5 was supported by the results of the LC-MS analysis of culture extracts, yielding very similar chromatographic profiles for both CDB and MEB culture extracts. As examined with reference to our in-house database and the MS-DIAL database, the monoisotopic mass data were clearly indicative of the presence of several typical *Fusarium* secondary metabolites (Table 4). In more detail, enniatins were consistently produced in both media; in fact, several members of this cyclohexadepsipeptide family were detectable in the analyzed culture extracts, with the notable exception of beauvericin. However, it is not possible to provide details on the identity for all of the analogues since some of them share the same molecular mass. The sesquiterpenoid culmorin was also detected in all of the analyzed samples, along with its precursor longiborneol. Likewise, the fusarielins, a series of polyketides with a decalin core, were also produced by both isolates; specifically, fusarielins A, B, F, and M were produced on both media, while other members of this family were detected only occasionally. Finally, chrysogine, a quinazolinone yellow pigment, was only detected in CDB cultures of both isolates.

**Table 4.** Secondary metabolites identified in culture filtrates of isolates Hzn1 and Hzn5.

| Compound | Formula | Monoisotopic Mass (MW) | Hzn1 | | Hzn5 | |
|---|---|---|---|---|---|---|
| | | | CDB | MEB | CDB | MEB |
| Enniatin A/C/F * | $C_{36}H_{63}N_3O_9$ | 681.456430 | + | + | + | + |
| Enniatin A1/E/G/I/O1/O2/O3 * | $C_{35}H_{61}N_3O_9$ | 667.440780 | + | + | + | + |
| Enniatin B | $C_{33}H_{57}N_3O_9$ | 639.409480 | + | + | + | + |
| Enniatin B1/B4/D/H * | $C_{34}H_{59}N_3O_9$ | 653.425130 | + | + | + | + |
| Enniatin B2/B3/J2/J3/K1 * | $C_{32}H_{55}N_3O_9$ | 625.393830 | + | + | + | + |
| Enniatin J1 | $C_{31}H_{53}N_3O_9$ | 611.378180 | + | + | + | + |
| Enniatin L/P1 * | $C_{34}H_{59}N_3O_{10}$ | 669.420045 | + | + | + | + |
| Enniatin P2 | $C_{33}H_{57}N_3O_{10}$ | 655.404395 | + | + | + | + |
| Chrysogine | $C_{10}H_{10}N_2O_2$ | 190.074227 | + | — | + | — |
| Fusarielin A | $C_{25}H_{38}O_4$ | 402.277009 | + | + | + | + |
| Fusarielin B | $C_{25}H_{40}O_5$ | 420.287574 | + | — | + | + |
| Fusarielin D/G * | $C_{25}H_{36}O_4$ | 400.261359 | — | — | + | — |
| Fusarielin E | $C_{25}H_{39}ClO_4$ | 438.253687 | + | — | — | — |
| Fusarielin F | $C_{25}H_{36}O_5$ | 416.256274 | — | + | + | + |
| Fusarielin M | $C_{25}H_{36}O_3$ | 384.266445 | + | + | + | + |
| Culmorin | $C_{15}H_{26}O_2$ | 238.193280 | + | + | + | + |
| Longiborneol | $C_{15}H_{26}O$ | 222.198365 | + | + | + | + |

* Metabolites with same MW and formula.

The analysis of the extract obtained from wheat plants at the end of the assays carried out on aphids showed the presence of a peak at 51.02 min, corresponding to enniatin B (mass of 639.4094). None of the other identified compounds could be detected in the plant extract.

## 4. Discussion

Members of the FCCSC have been described since 2018; in particular, the founding species *F. citricola* was identified along with *F. salinense* from symptomatic tissues of several *Citrus* species sampled in southern Italy [13]. Afterwards, a single strain of *F. celtidicola* was recovered from a dead branch of the lote tree (*Celtis australis*) in Italy [39], while *F. aconidiale* and *F. juglandicola* were, respectively, identified from wheat and from walnut (*Juglans regia*) buds and fruits in France [40,41]. Both of the latter species have been more recently detected on larvae and inside galls of the cecidomyid midges *Asphondylia echii* and *Lasioptera rubi* in Slovakia [42], while *F. juglandicola* has been identified in leaves of mistletoe (*Viscum album* subsp. *austriacum*) in Poland [43]. However, as usually happens with recently characterized taxa, the FCCSC could be more widespread and include isolates previously identified with other names. This is more than a mere hypothesis, since our phylogenetic analysis demonstrated that two isolates provisionally classified as *Fusarium* sp. clearly belong to this species complex: one of them was identifiable as *F. aconidiale*, while the other, positioned next to *F. salinense*, could be representative of a new species. All members of the FCCSC were confirmed to be related to *F. tricinctum*, while a higher phylogenetic distance was found with *F. lateritium*, consistent with the inference of a recent study by Turco and coworkers [17] that hazelnut isolates ascribed to the latter species may have been misidentified. Moreover, considering the available GenBank sequences, close matches were found with isolates collected from dead branches of sycamore maple (*Acer pseudoplatanus*) in Slovenia [44], from olives in Greece [45], and from the boxelder maple (*Acer negundo*) in Pomerania, northern Poland [46], which were all identified as *F. lateritium* based on an incomplete set of markers. In particular, as compared with our Hzn5, the strain FI47S-18An from the latter source displayed 99.24% homology with *tef-1* and 100% homology with *tub* sequences (Table 3); unfortunately, the *rpb2* sequence of this strain is not available, preventing its inclusion in our phylogenetic analysis. Interestingly, the above points are indicative of the widespread distribution in central and southern Europe of species in the FCCSC, calling for a revision of the taxonomic identification of isolates from this geographical area which were ascribed to *F. lateritium* before 2018.

The laboratory assays displayed antibiosis against plant pathogenic fungi and entomopathogenic effects against aphids, which were consistent with the production of bioactive secondary metabolites by our isolates. Anti-insect assays were carried out on a model aphid species available at our laboratory, the results of which could be reasonably transposed at least to other hemipterans infesting hazelnut. Likewise, the plant pathogens used for the dual-culture assays are representative of several Ascomycetes classes, depicting a broad range of antifungal properties. Of course, the ecological conditions can be very variable due to the unpredictable influence of many abiotic and biotic factors; nevertheless, these biological properties may be relevant for conjecturing a symbiotic association with hazelnut contributing to defensive mutualism.

The metabolomic profiles of our isolates were consistent in comparison with one another on both substrates used. Due to their recent identification, no data are currently available on the secondary metabolite profile of species belonging to the FCCSC, while those of the closely related FTSC have been accurately examined [47,48]. Among the products which best characterize the latter species complex, no clues could be detected for moniliformin, aurofusarin, chlamydosporol, acuminatopyrone, 2-amino-14,16-dimethyloctadecan-3-ol, and fungerin, while clear indications were obtained for the presence of enniatins. Recently, concern has been raised regarding food contamination by these products due to their cytotoxicity observed in several mammalian cell lines and possible synergic effects with other mycotoxins [49–52]. So far, the ability to produce enniatins has been reported in many *Fusarium* species of phytopathological relevance, upon both direct detection [53–57] and identification of biosynthetic genes [58–60]. A recent review mentions 28 species from 7 species complexes: in particular, *F. acuminatum*, *F. arthrosporioides*, *F. avenaceum*, *F. torulosum*, and *F. tricinctum* from the FTSC; *F. compactum*, *F. kyushuense*, *F. poae*, *F. sambucinum*, and *F. venenatum* from the *F. sambucinum* s.c.; *F. sporotrichioides* from the *F. sporotrichioides* s.c.; *F. acutatum*, *F. ananatum*, *F. andiyazi*, *F. antophilum*, *F. concentricum*, *F. fujikuroi*, *F. lactis*, *F. nygamai*, *F. proliferatum*, *F. ramigenum*, *F. subglutinans*, *F. temperatum*, *F. thapsinum*, and *F. verticillioides* from the *F. fujikuroi* s.c.; *F. equiseti* from the *F. incarnatum-equiseti* s.c.; *F. oxysporum* from the *F. oxysporum* s.c.; and *F. lateritium* from the FLSC [61]. Besides plant pathogens, the production of enniatins has been repeatedly reported in endophytic isolates, belonging to species such as *F. acuminatum* [62,63], *F. dimerum* [64], emphF. tricinctum, and related species in the FTSC [56,57,65–67], and to the *F. redolens* s.c. [68]. Moreover, these compounds have been characterized for their insecticidal activity for a long time [69,70], as well as for their antifungal and antibacterial properties [57,71,72], implying a possible role in defensive mutualism of plant endophytic strains. Notably, the detection of enniatin B in extracts from wheat plants treated with the Hzn5 conidial suspension is indicative that at least this compound can be produced in vivo and eventually exert its bioactivity against plant pests and pathogens.

First identified as a secondary metabolite of *Penicillium chrysogenum* [73], chrysogine is also widespread in *Fusarium*. In fact, this compound has been reported from *F. sambucinum* [74], *F. culmorum*, *F. equiseti* [75], *F. langsethiae*, *F. sporotrichioides* [76], *F. avenaceum* [77], *F. graminearum*, *F. pseudograminearum* [78], *F. tricinctum* [79], *F. cerealis* [80], and *F. oxysporum* [81], and more recently from an endophytic *F. multiceps* [82]. Moreover, the presence of the putative gene cluster for the biosynthesis of chrysogine has been documented in the new species *F. indicum* [83].

Conversely, fusarielins have been reported from a restricted range of *Fusarium* species so far. In fact, an investigation involving 42 strains from 18 species showed that these compounds could only be detected in cultures of *F. tricinctum* (fusarielin A) and *F. graminearum* (fusarielins F, G, and H) [84]. Besides fusarielin A, the production of fusarielins B, J, K, and L was later documented in endophytic isolates of *F. tricinctum* [57,85], while fusarielins M and N have been identified as products of marine strains of *F. graminearum* [86,87]; a few more reports refer to *Fusarium* isolates of unidentified species [88–90]. Hence, our isolates seem to combine the biosynthetic abilities of both the known producer species. In terms of biological properties, fusarielins A and E were first reported for their antifungal activities [88,89], while their antibacterial properties were documented for other

fusarielins [57,90], leaning toward an involvement of these compounds in the ecological interactions with other species which are part of the plant microbiome. Interestingly, genes for the biosynthesis of fusarielins were downregulated in the interaction of *F. graminearum* with the mycoparasite *Clonostachys rosea* [91]. Moreover, fusarielins have been reported to have antiproliferative [92] and estrogenic activities [93], calling for a more accurate assessment of their effects on human health.

First isolated from *F. culmorum* [94], culmorin is biosynthesized through the hydroxylation of longiborneol [95]. The latter compound, also produced by our isolates, is a known metabolite of *F. tricinctum* [48], while culmorin has never been reported from isolates of this species, despite a couple of biosynthetic genes being found in its genome [96]. Culmorin is regarded as an emerging mycotoxin, also produced by *F. graminearum*, *F. crookwellense*, and *F. venenatum* [97,98] where it co-occurs with trichothecenes, somehow influencing their toxicological properties [96,99]. However, no known trichothecenes could be identified in the secondary metabolite profiles of our strains. Similarly to fusarielins, moderate antifungal activity has been documented for culmorin [100], along with low insecticidal effects [101].

In addition to enniatins, it is relevant to consider that chrysogine [102,103], culmorin [97], and fusarielin A [104] have all been reported as contaminants in foodstuffs, calling for a more accurate assessment of their possible effects on human health, alone and in association with other co-occurring mycotoxins.

## 5. Conclusions

Within the context of uncertainty regarding the identification and the ecological role of *Fusarium* strains associated with hazelnut, two endophytic isolates collected in Poland were determined to belong to the FCCSC. This finding introduces the opportunity for a comparison with the hazelnut pathogenic strains previously identified as *F. lateritium*, also with the aim to assess whether the pathogenic aptitude on fruits is secondary to a more general endophytic habit.

Furthermore, this study represents the first metabolomic characterization of a member of the FCCSC, disclosing relatedness with the biosynthetic capabilities of species in the FTSC. The identified compounds, especially members of the enniatin family, may have implications in the defensive mutualism established in hazelnut plants by endophytic strains possessing these biosynthetic abilities; however, the infection of kernels progressing along with endophytic development may also affect the quality and safety of both fresh and processed hazelnut products.

**Author Contributions:** Conceptualization, B.Z. and R.N.; methodology, B.Z., A.L., G.M., K.W., I.K., A.S. and F.V.; validation, A.L., G.M. and F.V.; formal analysis, G.M., K.W., I.K. and A.S.; resources, B.Z., A.L., F.V. and R.N.; writing—original draft preparation, B.Z., A.L., G.M., F.V. and R.N.; writing—review and editing, B.Z., G.M. and R.N.; supervision, B.Z., F.V. and R.N.; funding acquisition, B.Z., F.V. and R.N. All authors have read and agreed to the published version of the manuscript.

**Funding:** This study was carried out within the Agritech National Research Center and received funding from European Union Next-Generation EU (Piano Nazionale di Ripresa e Resilienza (PNRR)—Missione 4 Componente 2, Investimento 1.4—D.D. 1032 17 June 2022, CN00000022). This manuscript reflects only the authors' views and opinions; neither the European Union nor the European Commission can be considered responsible for them. G.M. acknowledges funding from PON R&I 2014–2020 (FSE REACT-EU) DM 1062 Azione IV.6 (CUP: E65F21003080003).

**Institutional Review Board Statement:** Not applicable.

**Data Availability Statement:** The sequences of DNA markers generated in this study have been deposited in the GenBank repository (www.ncbi.nlm.nih.gov/genbank/ accessed on 2 May 2024).

**Acknowledgments:** The authors gratefully acknowledge the contribution of Simon Gibbons (Natural and Medical Sciences Research Center, University of Nizwa, Oman) for the revision of the English style of the final version of the manuscript.

**Conflicts of Interest:** The authors declare no conflicts of interest.

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
