# Peer review of "Fusarium and Hazelnut: A Story of Twists and Turns"

_agriculture, doi:10.3390/agriculture14071080_

Round 1
Reviewer 1 Report
Comments and Suggestions for Authors
The manuscript is written in a very pompous way. Many data from the general literature are being copied here and the list of references is enormous. More than 100 references covering almost 4 pages or ~1/4 of the entire document! It seems the authors needed to compensate for the minute amount of data they present. Moreover some of the literature data are presented as if the originate from the authors, e.g. lines 240-244.
Furthermore the data are incomplete (why is tub missing for Hzn1) and what are the quantitative data for the secondary metabolites (SM). Are we not merely discussing the presence for trace amounts?
If you work with strains isolated from hazelnut, why is the SM production than done with wheat plants?
Several annoying errors/typo’s are highlighted in the attached file, e.g. why is malt extract broth written as MEB and as MDB?

Comments on the Quality of English Languagesee above
Author Response
Thank you for your comments. Please, find below our point-by-point reply.
The manuscript is written in a very pompous way. Many data from the general
literature are being copied here and the list of references is enormous.
More than 100 references covering almost 4 pages or ~1/4 of the entire
document! It seems the authors needed to compensate for the minute amount
of data they present.
We believe that carefully taking into account the pertinent literature
positively qualifies a scientific paper. In our case we had to examine
several basic aspects, such as the previous reports from hazelnut, the
endophytic occurrence and mycotoxin production in Fusarium, which justify
the less than enormous list of references. The space covered by the list of
references at the end of the paper is due to the citation style of MDPI
journals, without any intent to compensate the amount of data.
Moreover some of the literature data are presented as if the originate from
the authors, e.g. lines 240-244.
Apart of just a single our previous work ([38]), we do not think to have
presented any other data from the literature as they originated from our own
work. As for lines 240-244, it is clearly stated that these are two
reference strains having the relevant DNA sequences deposited in GenBank
(hence, not ours!), which have been considered in the phylogenetic analysis.
Furthermore the data are incomplete (why is tub missing for Hzn1) and what
are the quantitative data for the secondary metabolites (SM). Are we not
merely discussing the presence for trace amounts?
Given the similarity between the two isolates, at first we only sequenced
one of them for species identification since the tubulin marker was previously used
for other Fusarium isolates from hazelnut. When we realized that this was
not sufficient, we shifted to sequencing the other markers of both Hzn1 and
Hzn5 to perform an accurate phylogenetic analysis. Of course, at this stage
the tub sequence of Hzn1 was not necessary anymore.
As for the SMs, we did not need to get quantitative data. In fact, the
amount of SMs produced in vitro largely depends on the culturing parameters,
and there is no correspondence with production in vivo. Our aim was just to
define the qualitative profile of these strains to point out possible
implications with both plant protection and mycotoxin contamination of
hazelnuts.
If you work with strains isolated from hazelnut, why is the SM production
than done with wheat plants?
We clearly explained that this model assay was carried out because of the
availability of R. padi bred on wheat plants at the laboratory of the
Department of Plant Protection in Lublin. Indeed, this assay was quite
useful since it disclosed the ability by our Fusarium strains to settle as
endophytes on a herbaceous plant where they even produced enniatin B, with
an ensuing detrimental effect on the aphid colony development.
Several annoying errors/typo’s are highlighted in the attached file, e.g.
why is malt extract broth written as MEB and as MDB?
All typos indicated in the attached file were corrected. Moreover, the
English style of the text was checked by an English mother tongue person.
Reviewer 2 Report
Comments and Suggestions for Authors
The authors need more isolates of Fusarium in the study. So far, it is hard to reach to a conclusion.
Author Response
Thank you for your comment. It is in line with our conclusion that isolations from other areas and comparison with the Italian strains will clarify both the taxonomic issues and the ecological role of Fusarium associates of hazelnut.
Reviewer 3 Report
Comments and Suggestions for Authors
The MS describes the strains isolated from hazelnut in a versatile way, so in addition to the traditional mycological description of the Fusarium species, the phylogenetic analysis and the profile of the secondary metabolites also play a role. The developed research plan, the methods used, and the evaluation of the results all reflect serious professional background knowledge. Defensive mutualism is a good approach to the possible ecological role of the investigated strains, which is also justified by the large number of bioactive substances.
The manuscript is readable, easy to understand, surprisingly, it does not even contain letter/character errors.
Hopefully, the response codes have already arrived from GenBank in the meantime. (Ad 227-229)
Comments on the Quality of English LanguageIt corresponds to the standard expected of a scientific paper.
Author Response
Thank you very much for your very positive comments and appreciation of our English style. However, following comments by reviewer #1, we had the text revised by an English mother tongue person. As for the GenBank codes, we register an unusual delay, since they have not been released yet; thus, we retain the supplementary file reporting them in full details. Nevertheless, we are confident that in future it will be not problematic for readers to access the GenBank files by searching through the isolate numbers.
Round 2
Reviewer 1 Report
Comments and Suggestions for Authors
The document has been improved by eliminating the indicated typo’s, but the rebuttal is insufficient to change my evaluation.
I stick to my criticisms on the document and suggest to reject the manuscript. Albeit of describing two strains of Fusarium from hazelnut, a under lighted field of research, the document does not add much to the scientific community. It is well-known that morphology and ITS sequencing are not very informative in Fusarium taxonomy, the MS deals in great length on these topics.
My remark on the length of the literature section cannot be blamed on the format of the journal. You just need to reduce the number of citations. This is supposed to be a scientific report and NOT a literature overview!
The sequences generated in this work are not disclosed which limits the potential for checking statements. Moreover the general policy on sequence data is to deposit them in recognised depositories prior to publication.
The absence of quantitative data on SM production makes me wonder if these are not just trace amounts and how can you rule out this is caused by spill-over from other experiments? The (qualitative) differences between Hzn1 and Hzn5 for several fusarielins (see table 4) adds to that thought.
Figure 1 has a weird appearance
Comments on the Quality of English LanguageThe document has been improved by eliminating the indicated typo’s, but it still is rather pompous. E.g why is <aptitude> replaces only once in the text but not in the abstract?
Author Response
The manuscript has been fixed following the Academic Editor's advice.
Reviewer 2 Report
Comments and Suggestions for Authors
No further advince
Author Response

(The authors gave the same response as above.)
